# CPMöbius: Iterative Coach–Player Reasoning for Data-Free Reinforcement Learning

**Ran Li** [* 1 2]  **Zeyuan Liu** [* 1]  **Yinghao Chen** [1]  **Bingxiang He** [1]  **Jiarui Yuan** [1]  **Zixuan Fu** [1]  **Weize Chen** [1]  **Jinyi Hu** [1]
**Chen Qian** [3]  **Zhiyuan Liu** [1]  **Maosong Sun** [1]

## Abstract

Large Language Models (LLMs) have demonstrated strong potential in complex reasoning, yet their progress remains fundamentally constrained by reliance on massive high-quality human-curated tasks and labels, either through supervised fine-tuning (SFT) or reinforcement learning (RL) on reasoning-specific data. This dependence renders supervision-heavy training paradigms increasingly unsustainable, with signs of diminishing scalability already evident in practice. To overcome this limitation, we introduce **CPMöbius**, a collaborative **Coach–Player** paradigm for data-free reinforcement learning of reasoning models. Unlike traditional adversarial self-play frameworks, **CPMöbius** inspired by multi-agent collaboration, treats the Coach and Player as independent but cooperative roles. The Coach proposes instructions targeted at the Player's capability and receives rewards based on changes in the Player's performance, while the Player is rewarded for solving the increasingly instructive tasks generated by the Coach. This cooperative optimization loop is designed to directly enhance the Player's mathematical reasoning ability. Remarkably, **CPMöbius** achieves substantial improvement without relying on any external training data, outperforming existing unsupervised approaches. For example, on Qwen2.5-Math-7B-Instruct, our method improves accuracy by overall average +4.9 and out-of-distribution average +5.4, which exceed RENT by +1.5 on overall accuracy and R-zero by +4.2 on OOD accuracy. Our codebase has been released at https://github.com/thunlp/CPMobius.

[1]Tsinghua University, [2]University of Cambridge [3]Shanghai Jiao Tong University. Correspondence to: Ran Li <rl810@cam.ac.uk>, Weize Chen <chenwz21@mails.tsinghua.edu.cn>, Zhiyuan Liu <liuzy@tsinghua.edu.cn>.

*Proceedings of the 43rd International Conference on Machine Learning*, Seoul, South Korea. PMLR 306, 2026. Copyright 2026 by the author(s).

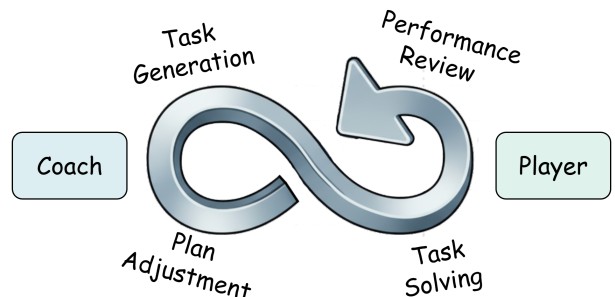

*Figure 1.* **CPMöbius** starts with the coach proposing tasks of suitable difficulty. The player learns by solving these tasks, then reviews on a predefined environment. Finally, the coach adjusts the next training plan based on the player's performance.

## 1. Introduction

Large Language Models (LLMs) (OpenAI, 2025a; Yang et al., 2024a; Touvron et al., 2023) have demonstrated remarkable capabilities in complex reasoning tasks, from mathematical reasoning, problem solving (Wei et al., 2022) to code generation (Chen et al., 2021). The dominant paradigm for enhancing these abilities involves post-training on domain-specific data, typically through supervised fine-tuning (SFT) (Ouyang et al., 2022; Tunstall et al., 2023) followed by reinforcement learning (RL) (Christiano et al., 2017; Schulman et al., 2017). While effective, these approaches are fundamentally constrained by their reliance on massive, high-quality, human-curated datasets. The scarcity of such expert-produced examples means this highly supervision-dependent paradigm is showing signs of strain, raising concerns about its long-term scalability.

To break free from this dependency, a promising frontier has emerged in data-free learning, where models improve through autonomous interaction. Self-play, a concept inspired by game-playing AI (Silver et al., 2017), has been adapted for LLM reasoning to achieve self-evolving. Recent self-play frameworks in RL (Huang et al., 2025; Zhao et al., 2025) have shown that a model can generate its own training data and learn from solving them, entirely removing the need for external datasets. However, these pioneering methods are often built on an adversarial or

competitive dynamic, where the model in one role generates challenges to stump another. Such an adversarial setup is prone to instability, collapsing into nonsensical or unlearnable proposed tasks for RL training.

In this work, we propose **CPMöbius**, a Coach–Player paradigm for data-free reinforcement learning, inspired by real world human sports collaboration and multi-agent collaboration (Chen et al., 2024; Qian & Cong, 2023). Instead of casting the Player model as competitors, the Coach is responsible for adapting the task difficulty to the Player's capabilities. **CPMöbius** treats the Coach and Player models as independent but collaborative partners in a symbiotic learning process. Throughout this paper, **"data-free"** refers only to the co-evolution stage after Coach–Player collaboration begins, and does not count any one-time model initialization performed beforehand. As shown in Fig. 1, the Coach and Player models are optimized through a cooperative loop:

- The **Coach** model acts as a curriculum designer, proposing maximally instructive tasks targeted at the Player's current capability.
- The **Player** model focuses on solving these tasks to enhance its reasoning skills.
- The **reward signals** for both Coach and Player are designed to foster cooperation. The Coach is rewarded based on the **environment feedback-induced accuracy gap of the Player**, directly incentivizing it to generate instructions that lead to tangible learning progress. Simultaneously, the Player is rewarded via a **standard verifiable outcome** for correctly solving tasks provided by the Coach.

A critical ingredient in this paradigm is a **Coach that can genuinely "teach"**. It must ask constructive, targeted questions and scaffold the Player with tasks that are informative rather than random. In practice, a weak or unskilled Coach tends to generate ambiguous or unhelpful tasks, which makes the feedback noisy and undermines co-evolution.

This collaborative dynamic allows **CPMöbius** to generate a highly targeted and adaptive curriculum from scratch, tailored specifically to the Player's evolving needs throughout the training process. Our experiments show that this data-free, cooperative approach is not only viable but remarkably effective. Without relying on any external training data during co-evolution, **CPMöbius** achieves substantial improvements and outperforms existing unsupervised methods. For instance, on the Qwen2.5-Math-7B-Instruct, our method improves accuracy by overall average +4.9 and out-of-distribution average +5.4, a significant leap compared to the +1.5 from RENT, a method of reinforcement learning via entropy minimization (Prabhudesai et al., 2025) and +4.2 from R-zero. The details of these baseline methods are provided in Section 5.1 These results demonstrate the effectiveness and scalability of collaborative paradigm as

a new pathway for advancing mathematical reasoning in LLMs, decoupling their progress from the constraints of human supervision.

## 2. Related Work

**Reinforcement Learning with Verifiable Rewards.** Recent advances in language model reasoning have leveraged Reinforcement Learning with Verifiable Rewards (RLVR), in which models are trained using binary feedback derived from programmatic verifiers that check correctness against ground truth (Lambert et al., 2024; Guo et al., 2025; Zhang et al., 2025a). By replacing learned reward models with rule-based verifiers, RLVR enables reliable optimization and mitigates reward hacking. Leading systems (Jaech et al., 2024; OpenAI, 2025b;a; Agarwal et al., 2025a; Comanici et al., 2025; Seed et al., 2025) demonstrate that RLVR can substantially improve reasoning and problem-solving abilities. Typical rule-based rewards include accuracy checks for deterministic outcomes and format constraints for structured outputs, both of which enhance the reliability and reproducibility of large-scale RL training pipelines. Despite their effectiveness, RLVR is fundamentally limited by the availability of verifiable supervision, which becomes increasingly costly as models surpass human-level expertise in specialized domains (Burns et al., 2023).

**Self-Play and Co-Evolving Policy-Rewards.** Self-play has emerged as a powerful paradigm for improving LLMs without relying solely on external supervision. In this approach, a model either generates its own training signals or interacts with a counterpart to refine both policy and reward (Yuan et al., 2024; Jiang et al., 2025). Techniques include self-rewarding, where a model critiques or corrects its own outputs (Xiong et al., 2025; Zhang et al., 2025b; Team, 2025), and co-optimization, where the policy and a separate reward model are trained jointly to enhance robustness and reduce reward hacking (Zha et al., 2025; Hong et al., 2025; Lu et al., 2025). By unifying the roles of generator and verifier, self-play enables dynamic adaptation and continuous improvement, offering a scalable alternative to purely supervised or static reward schemes.

**Data-Free Reinforcement Learning.** To address the limitations of human-generated rewards, recent work has explored data-free RL methods that generate training signals automatically. Some approaches leverage a model's own outputs or internal states, using consistency, confidence, or self-evaluation to guide learning (Zuo et al., 2025; Agarwal et al., 2025b; Li et al., 2025; Yuan et al., 2024). Others rely on external, automated signals, such as heuristics or the structure of large unlabeled corpora (Dong et al., 2025; Zweiger et al., 2025). More sophisticated methods combine these ideas, allowing models to generate problems for

themselves, evaluate solutions, and iteratively refine both policy and reward (Zhao et al., 2025; Huang et al., 2025; Chen et al., 2025; He et al., 2026). Together, these data-free approaches provide scalable training for LLMs, enabling self-improvement without human labels, though they remain sensitive to reward misalignment and can exhibit failure modes such as collapse or repetitive behavior.

# 3. Preliminaries

In this section, we briefly review two key RL methods for LLM that are relevant to our framework.

## 3.1. Group Relative Policy Optimization

Group Relative Policy Optimization (GRPO) (Shao et al., 2024), proposed by DeepSeek, is a *critic-free* reinforcement learning algorithm. Given a query $q$, GRPO samples $G$ candidate outputs $\{o_1, \ldots, o_G\}$ from the old policy $\pi_{\theta_{\text{old}}}$, and defines the normalized advantage function using the corresponding rewards $\{r_1, \ldots, r_G\}$:

$$A_i = \frac{r_i - \text{mean}(\{r_1, r_2, \ldots, r_G\})}{\text{std}(\{r_1, r_2, \ldots, r_G\})} \quad (1)$$

The policy $\pi_\theta$ is then updated by maximizing the following objective:

$$J_{\text{GRPO}}(\theta) = \mathbb{E}_{q, \{o_i\}} \left[ \frac{1}{G} \sum_{i=1}^{G} \min \left( r_i(\theta) A_i, \right. \right.$$

$$\left. \left. \text{clip}(r_i(\theta), 1 - \epsilon, 1 + \epsilon) A_i \right) \right] - \beta D_{\text{KL}}(\pi_\theta \| \pi_{\text{ref}}) \quad (2)$$

where $\epsilon$ and $\beta$ are hyperparameters, $r_i(\theta) = \frac{\pi_\theta(o_i|q)}{\pi_{\theta_{\text{old}}}(o_i|q)}$ is the importance sampling ratio, and $D_{\text{KL}}(\pi_\theta \| \pi_{\text{ref}})$ is the KL divergence regularization with respect to a reference model.

## 3.2. Reinforcement Learning with Verifiable Rewards

Reinforcement Learning with Verifiable Rewards (RLVR) is a framework that trains models using verifiable reward functions without relying on human feedback (Lambert et al., 2024). In RLVR, the reward function is typically defined by deterministic rules that automatically assess the correctness of model outputs, providing binary signals (1 for correct, 0 for incorrect):

$$r(y) = \texttt{verify}(y), \quad (3)$$

where $\texttt{verify}(\cdot)$ is a verifiable function determining whether the output $y$ is correct.

Depending on the verification source, rewards can be obtained in different ways. When ground truth labels $y^\star$ are available, the accuracy is determined by direct comparison $r(y) = \mathbf{1}[y = y^\star]$, as in Group Relative Policy Optimization (GRPO), where rules-based rewards check both the accuracy of the solutions and the required output format. In the absence of labels, verification can be performed in an unsupervised manner using self-consistency (Wang et al., 2023; Zuo et al., 2025), where the majority-voted answer from a set of candidate outputs is treated as the correct answer and rewards are assigned accordingly. This formulation highlights that verifiable rewards can be constructed either with or without supervision, enabling reinforcement learning to be applied even in data-scarce or fully unsupervised reasoning scenarios.

# 4. Framework

In this section, we present a comprehensive overview of **CPMöbius**, a collaborative Coach–Player paradigm for data-free reinforcement learning. **CPMöbius** introduces a symbiotic learning loop between two independent language models: the **Coach**, a curriculum designer, and the **Player**, a reasoning solver.

The core objective is to maximize learning progress without human-curated data. To achieve this, the Coach generates mathematical tasks tailored to the Player's current capability, while the Player attempts to solve them. The key innovation lies in the cooperative reward mechanism: the Coach is optimized not to stump the Player, but to maximize the Player's capability based on Coach-proposed tasks. This ensures that the curriculum remains instructive, learnable, and adaptive.

We illustrate the main framework in Fig. 2, and the pseudo-code of algorithm can be found in Appendix A.1. Formally, let $\pi_\theta^{\text{C}}$ denote the Coach policy and $\pi_\phi^{\text{P}}$ the Player policy. At each round $t$:

1. **Coach designs plan.** The Coach generates a batch of $m$ task instructions $\{x_i\}_{i=1}^m \sim \pi_{\theta_t}^{\text{C}}(\cdot)$, where $\pi_{\theta_t}^{\text{C}}$ is the current Coach policy.

2. **Player executes training.** For every $x_i$ the current Player produces $n$ independent answers $\{y_{i,j}\}_{j=1}^n \sim \pi_{\phi_t}^{\text{P}}(\cdot|x_i)$. Majority voting over the $n$ answers yields a *pseudo-label* $y_i^*$. Then each answer receives a verifiable reward $r_{i,j} = \mathbb{I}[y_{i,j} = y_i^*]$ as well as a GRPO advantage $A_{i,j}$ computed w.r.t. the $n$ samples for question $i$. The *instruction-level training reward* is obtained by averaging: $R_i^{\text{Player}} = \frac{1}{n} \sum_{j=1}^n r_{i,j}$. The set $\{(x_i, \{y_{i,j}\}_{j=1}^n)\}_{i=1}^m$ constitutes one GRPO batch, and Player parameters $\phi_t$ are updated using GRPO method while keeping KL within a trust-region.

3. **Player evaluates results.** The updated Player receives *environment feedback* (computed using a fixed evaluation

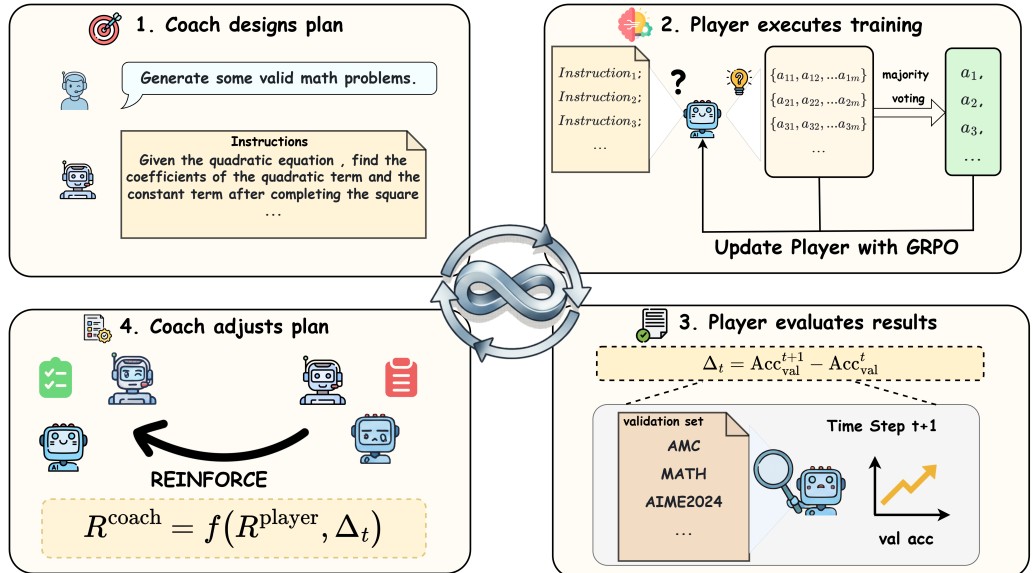

*Figure 2.* The illustration on the conceptual layered architecture on the design of **CPMöbius**. The iterative process includes four stages. **Coach designs plan**: The coach gives instructions of suitable difficulty based on the player's current ability. **Player executes training**: The player executes each instruction multiple times, uses majority voting to get pseudo-labels, and updates with GRPO. **Player evaluates results**: The updated player is interacted on a prepared environment, and the environment feedback-induced accuracy gap is recorded. **Coach adjusts plan**: The coach updates with REINFORCE, using the player's performance on both the proposed instructions and the environment feedback as rewards.

set $\mathcal{D}_{\text{val}}$), yielding a *progress reward*

$$\Delta_t = \text{Acc}\left(\pi^{\text{P}}_{\phi_{t+1}}; \mathcal{D}_{\text{val}}\right) - \text{Acc}\left(\pi^{\text{P}}_{\phi_t}; \mathcal{D}_{\text{val}}\right),$$

which measures the Player's accuracy difference after receiving environment feedback.

4. **Coach adjusts plan.** Each instruction $x_i$ is assigned an *instruction reward* $R_i^{\text{Coach}} = R_i^{\text{Player}} \cdot \Delta_t$, i.e., instructions that produced high Player rewards and coincided with a global accuracy improvement are reinforced. A group of $m$ instruction-level REINFORCE steps update Coach parameters $\theta_t$ using each instance in the batch $\{(x_i, R_i^{\text{Coach}})\}_{i=1}^m$.

The entire loop is trained end-to-end with separate policy optimization for Coach and Player using the REINFORCE and GRPO, respectively. Critically, no human prompts and no external curricula are ever used. The Coach learns to teach, and the Player learns to solve, purely through interaction with each other. This cooperative design sidesteps the instability of adversarial self-play while retaining the benefits of open-ended, adaptive curriculum generation. In the following subsections, we detail the architecture, reward design, and training procedure of both the Coach and the Player.

## 4.1. Coach

The Coach serves as an adaptive *curriculum designer*, fundamentally responsible for generating tasks that improve the Player's current reasoning capabilities. Unlike traditional static curriculum approaches, our Coach acts as a dynamic learning policy that continually refines its task-generation strategy in response to the Player's learning trajectory. **The Coach never observes ground-truth solutions**; instead, it receives only a scalar *environment feedback* signal, $\Delta_t$, which captures the post-update performance change (computed using a fixed evaluation set $\mathcal{D}_{\text{val}}$).

**Difficulty-Filtered Batching** To ensure that every proposal task is *learnable yet non-trivial*, we use a lightweight difficulty check during the task-generation phase. For each candidate task $x_i$ sampled from $\pi_\theta^{\text{C}}$, we rollout $n$ Player answers $\{y_{i,j}\}_{j=1}^n \sim \pi_\phi^{\text{P}}(\cdot|x_i)$, obtain the majority-voted pseudo-label $y_i^*$, and compute the rollout-dependent accuracy score of the instruction.

$$acc_i = \frac{1}{n} \sum_{j=1}^n \mathbb{I}[y_{i,j} = y_i^*]. \tag{4}$$

This score effectively measures the problem's alignment with the Player's current capability. The Coach then applies a principled filtering criterion, retaining only problems whose accuracy scores fall within the pedagogically optimal zone of $0.2 \leq acc_i \leq 0.8$. Problems outside this

range are immediately discarded and replaced through on-the-fly resampling. This online filter guaranties that the final mini-batch of $m$ questions is challenging enough to promote skill development yet solvable enough to avoid frustration, providing a natural curriculum ramp.

**Design Objectives** The Coach embodies a learner-centered educational philosophy, where its primary objective is to optimize the constructiveness of the proposed-task for the Player. Formally, the Coach policy $\pi_\theta^{\text{C}}$ is optimized using instruction-level rewards that combine local training effectiveness with global educational outcomes.

$$R_i^{\text{Coach}} = R_i^{\text{Player}} \cdot \Delta_t \tag{5}$$

where

$$R_i^{\text{Player}} = \frac{1}{n} \sum_{j=1}^{n} r_{i,j} \tag{6}$$

represents the average training reward achieved by the Player on instruction $x_i$, and $\Delta_t = \text{Acc}_{\text{val}}(\pi_{\phi_{t+1}}^{\text{P}}) - \text{Acc}_{\text{val}}(\pi_{\phi_t}^{\text{P}})$ measures the Player's accuracy improvement after receiving environment feedback

This multiplicative reward embodies a pedagogical principle: proposed tasks receive positive reinforcement only when they **simultaneously achieve high Player performance during training (high $R_i^{\text{Player}}$) and contribute to measurable learning progress (positive $\Delta_t$)**. The Coach parameters are updated through REINFORCE using the batch of instruction-reward pairs $\{(x_i, R_i^{\text{Coach}})\}_{i=1}^{m}$:

$$\nabla_\theta J(\theta) = \frac{1}{m} \sum_{i=1}^{m} R_i^{\text{Coach}} \nabla_\theta \log \pi_\theta^{\text{C}}(x_i). \tag{7}$$

### 4.2. Player

The Player functions as the primary reasoning model, designed to develop robust mathematical problem-solving capabilities through iterative interaction with the Coach-generated curriculum.

**Design Objectives and Collaborative Dynamics** The Player's core objective is to maximize solving accuracy on mathematical problems while developing generalizable reasoning strategies. The Player operates within a collaborative learning framework where its performance directly influences curriculum adaptation through a sophisticated feedback mechanism. The Player's learning process is also inherently adaptive, continuously calibrating its problem-solving strategies based on feedback from the Coach-generated curriculum.

The Player employs multi-sample reasoning for each problem $x_i$, generating $n$ independent solution attempts $\{y_{i,j}\}_{j=1}^{n}$ to enable robust pseudo-label generation through

majority voting. This approach mitigates individual reasoning errors, provides confidence estimates for generated solutions, and creates multiple learning signals from each instructional instance.

The interaction protocol between the Coach and Player establishes a dynamic feedback loop that drives mutual improvement. This ensures the curriculum remains at an optimal difficulty, maintaining learning momentum and continuously pushing the frontier of the Player's capabilities.

**Training and Optimization** The Player is optimized using GRPO, which enables stable learning from the pseudo-labels generated through majority voting. For each problem instance $x_i$, the Player receives rewards

$$r_{i,j} = \mathbb{I}[y_{i,j} = y_i^*], \tag{8}$$

where $y_i^*$ is the majority-voted pseudo-label. The GRPO advantage computation considers the relative performance across the $n$ samples for each problem:

$$A_{i,j} = \frac{r_{i,j} - \text{mean}(\{r_{i,1}, r_{i,2}, \ldots, r_{i,n}\})}{\text{std}(\{r_{i,1}, r_{i,2}, \ldots, r_{i,n}\})} \tag{9}$$

This collaborative process completes the **CPMöbius** training loop: the Coach designs training curriculum, the Player explores potential solutions, and the Player's consequent capability guides the curriculum's evolution. The process is inherently curriculum-aware, prioritizing challenging yet solvable problems to ensure the Player's skill development remains aligned with the Coach's adaptive strategy. Through this orchestrated interaction, the framework achieves data-free mathematical reasoning development, where both models co-evolve to maximize learning efficiency without reliance on human-curated data or a pre-defined curriculum.

## 5. Experiments

### 5.1. Experiment Setup

**Coach Model Selection.** We fix the Coach to **Qwen2.5-Math-7B-Instruct** (**Yang et al., 2024b**) that is further **warmed up with 4K PRIME Eurus-2-RL-Data** (**Cui et al., 2025**). This warm-up is mainly used to strengthen the Coach's ability to pose constructive, diagnostically useful questions, which in turn improves the quality of the curriculum it generates and facilitates effective co-evolution with the Player. Importantly, our use of warm-up data does not contradict the "no external training data" setting in the co-evolution stage. No additional external training data is introduced *after* the warm-up, and all subsequent learning signals arise from the Coach and Player interaction and the environment feedback. Our setting is ***data-free Player training with an optionally warmed-up Coach***.

*Table 1.* Performance comparison between **CPMöbius** and baseline methods on mathematical reasoning benchmarks. Overall Average indicates the mean performance over all benchmarks. OOD Average refers to the out-of-distribution performance, computed as the mean across all benchmarks except the AMC datasets, because RENT was trained on AMC and **CPMöbius** validation also used AMC. This separation enables a fair comparison by clearly distinguishing in-distribution (AMC) results from out-of-distribution generalization performance. Bold values indicate best performance for each metric.

| Models | Average | OOD Average | AMC | AIME 2024 | AIME 2025 | Minerva | MATH | Olympiad |
|---|---|---|---|---|---|---|---|---|
| *Qwen2.5-Math-1.5B* | | | | | | | | |
| Base Model | 23.3 | 19.8 | 34.6 | 6.2 | 2.8 | 16.3 | 56.2 | 23.4 |
| R-Zero (Iter 3) | 27.1 | 24.7 | 39.2 | 9.8 | 5.0 | 19.3 | 62.4 | 26.8 |
| RENT | 27.1 | 24.7 | 39.3 | **10.0** | 5.0 | 19.0 | 62.2 | **27.1** |
| *CPMöbius* | **28.8** | **26.8** | **39.4** | 9.8 | **5.4** | **28.0** | **63.1** | 26.9 |
| *OpenMath-Nemotron-1.5B* | | | | | | | | |
| Base Model | 59.5 | 54.9 | 82.3 | **55.6** | 43.3 | **25.1** | 89.4 | 61.0 |
| R-Zero (Iter 3) | - | - | - | - | - | - | - | - |
| RENT | 61.7 | 56.5 | **87.7** | 55.0 | 46.0 | 24.2 | 90.7 | 66.7 |
| *CPMöbius* | **62.1** | **57.0** | 87.5 | 54.9 | **46.9** | 24.3 | **91.2** | **67.9** |
| *OctoThinker-3B-Hybrid-Zero* | | | | | | | | |
| Base Model | 21.3 | 20.6 | 24.6 | 3.9 | 1.7 | 16.3 | 57.9 | 23.4 |
| R-Zero (Iter 3) | 20.5 | 19.5 | 25.9 | 2.0 | 0.3 | 14.6 | 58.1 | 22.3 |
| RENT | 23.0 | 21.7 | **29.2** | **7.3** | **2.1** | 15.0 | 60.2 | 24.1 |
| *CPMöbius* | **23.6** | **22.0** | 28.0 | 4.8 | 1.7 | **22.1** | **60.4** | **24.7** |
| *Qwen2.5-Math-7B-Instruct* | | | | | | | | |
| Base Model | 35.8 | 33.0 | 49.2 | 9.0 | 6.3 | 34.6 | 78.0 | 37.4 |
| R-Zero (Iter 3) | 36.9 | 34.2 | 50.5 | 9.5 | 7.4 | 32.7 | 83.3 | 38.1 |
| RENT | 39.2 | 37.6 | 53.1 | 10.8 | **9.9** | 38.8 | 83.8 | **38.8** |
| *CPMöbius* | **40.7** | **38.4** | **55.6** | **11.8** | 9.6 | **44.9** | **84.2** | 38.3 |

**Player Model Selection.** We select four base models for our training experiments, representing the three main stages of a typical LLM training lifecycle: pre-training, supervised fine-tuning (SFT), and reinforcement learning.

- **Qwen2.5-Math-1.5B** (Yang et al., 2024b): a mathematical pre-training model.
- **OpenMath-Nemotron-1.5B** (Moshkov et al., 2025): a large-scale SFT enhanced model based on Qwen2.5-Math-1.5B.
- **Qwen2.5-Math-7B-Instruct** (Yang et al., 2024b) and **OctoThinker-3B-Hybrid-Zero** (Wang et al., 2025): models optimized through reinforcement learning.

More details about these models are introduced in Appendix A.2.

**Training Details.** All experiments were conducted within the verl (Sheng et al., 2025). We use AMC as the fixed held-out validation $D_{val}$ during training. We choose AMC because its difficulty is typically moderate, it is neither trivial which would quickly saturate and yield a near-zero learning signal nor overly hard which would produce sparse, noisy feedback early on. As a result, AMC provides a more stable and informative progress signal for computing the environment feedback $\Delta_t$ throughout training. All experiments were conducted using 4 to 8 NVIDIA A800-80GB GPUs per setting. We set the batch

size as 16 and the number of rollout samples for each prompt as 16, ensuring that each training round involves the Coach generating 16 questions and the Player producing 16 candidate solutions for majority voting-based pseudo-label generation. More hyperparameter configurations and prompt templates are provided in Appendix A.6.

**Evaluation Details.** We evaluate the Player models on six established mathematical reasoning benchmarks spanning diverse difficulty levels: AMC, Minerva (Lewkowycz et al., 2022), MATH-500 (Hendrycks et al., 2021), Olympiad-Bench (He et al., 2024), and AIME 2024 and AIME 2025. To ensure robustness, we employ benchmark-specific sampling strategies calibrated to each benchmark's difficulty: mean@32 for AIME benchmarks, mean@10 for AMC, mean@6 for Minerva, mean@5 for MATH-500, and mean@3 for Olympiad-Bench. Since AMC is used as the validation set during training, we compute both the average score on all six datasets and the OOD average score on the other five datasets except for AMC. All sampling settings are kept consistent with the training configuration, as illustrated in Appendix A.6.

**Baselines.** For our main experiments, beyond the selected base models, we considered two representative unsupervised training paradigms as baselines. The first is RENT (Prabhudesai et al., 2025), which employs entropy minimization: the model's own confidence in its generated answers is

treated as a reward signal, without relying on external feedback. The second is R-Zero (Huang et al., 2025), which initializes two roles of the same model that interact adversarially, with the challenger generating tasks and the solver attempting to solve them.

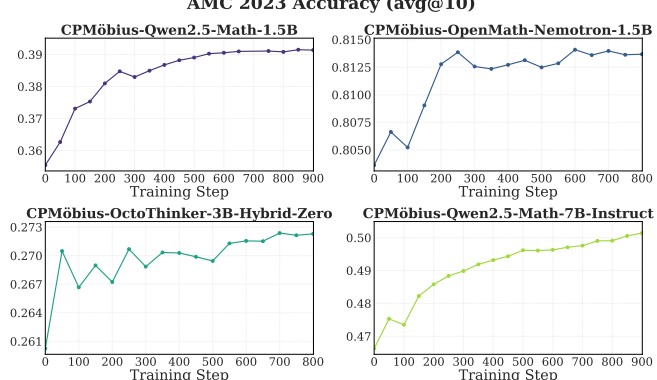

*Figure 4.* Visualization of the training dynamics of **CPMöbius** using validation results on AMC dataset. The curves are smoothed with Time Weighted EMA, where **CPMöbius** shows consistent performance improvement for different base models.

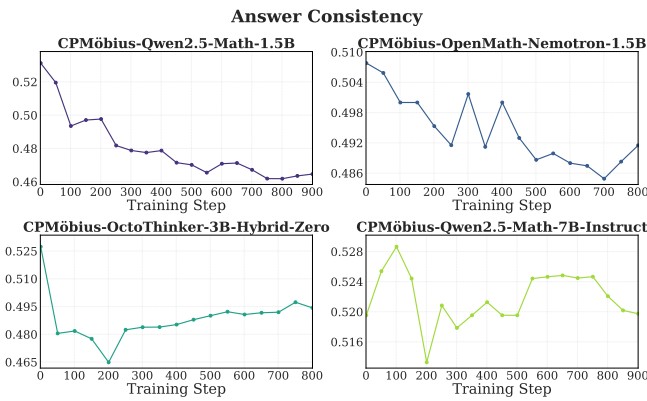

*Figure 3.* Visualization of the Player's answer consistency on Coach proposed tasks during training. A lower value indicates higher difficulty of the instructions.

## 5.2. Results

We present the main results in Table 1. We have the following observations:

**CPMöbius outperforms other unsupervised RL methods:** The results show that **CPMöbius** achieves superior performance compared to other unsupervised RL baselines, consistently achieving the highest overall average and OOD average scores across all four base model. Impressively, **CPMöbius** successfully improves high-performing base models OpenMath-Nemotron-1.5B (from 59.5 to 62.1). Notably, we found that the method from R-Zero failed on OpenMath-Nemotron-1.5B, failing to be trained as a Challenger as required by R-Zero. This demonstrates **CPMöbius**'s ability to push models beyond their apparent performance ceiling, a critical advantage for practical applications where starting from pre-optimized models is common.

**Strong out-of-distribution generalization:** **CPMöbius** achieves better OOD average scores across all four tested models, demonstrating that the reasoning capabilities learned from AMC competition problems effectively transfer to diverse mathematical domains. On MATH, **CPMöbius** consistently outperforms other methods with improvements ranging from 1.8 to 6.9 points over base models. The most striking OOD generalization occurs on the Minerva benchmark, where **CPMöbius** achieves obvious improvements: from 16.3 to 28.0 (71.8%) on Qwen2.5-Math-1.5B and 34.6 to 44.9 (29.8%) on Qwen2.5-Math-7B-Instruct.

**Performance analysis for different initial models:** The experimental results reveal distinct performance patterns that correlate with initial model characteristics. (1) Foundation models demonstrate high improvement potential: Qwen2.5-Math-1.5B achieves an overall 5.5 points improvement (23.6% relative gain), suggesting that models with domain-specific pre-training provide strong foundations for **CPMöbius**'s optimization approach. (2) SFT-enhanced models show diminishing but meaningful returns: Despite starting from a high 59.5 points baseline after extensive SFT on 5.5 million instances, OpenMath-Nemotron-1.5B still achieves an overall 2.6 points improvement, demonstrating **CPMöbius**'s ability to push beyond traditional SFT limits. (3) RL-optimized models exhibit varied enhancement: Qwen2.5-Math-7B-Instruct shows remarkable 4.9 points improvement despite instruction tuning, while OctoThinker-3B-Hybrid-Zero shows modest 2.3 points gains.

**Robustness to different held-out environments.** To examine whether the effectiveness of **CPMöbius** depends on the choice of the held-out validation environment, we further replace AMC with Minerva and OlympiadBench as the validation environment for computing the Coach reward signal. As shown in Appendix A.5, **CPMöbius** still brings consistent improvements on both Qwen2.5-Math-1.5B and Qwen2.5-Math-7B-Instruct. These results suggest that the proposed Coach–Player paradigm is not merely exploiting dataset-specific patterns from AMC, but can generalize across different validation environments and continue to improve the Player's reasoning ability.

## 5.3. Training Dynamics

We analyze the training dynamics of **CPMöbius** by tracking both validation accuracy on AMC and the consistency of the Player's responses throughout training steps. As shown

*Table 2.* Ablation study results are based on the Qwen2.5-Math-1.5B base model. **w/o Coach Update**: disables training of the Coach. **w/o Coach Warm-up**: uses the base model as the Coach. **w/o Instruction Filter**: disables difficulty filtering by the Coach.

| Models | Average | OOD Average | AMC | AIME 2024 | AIME 2025 | Minerva | MATH | Olympiad |
|---|---|---|---|---|---|---|---|---|
| *Qwen2.5-Math-1.5B* | | | | | | | | |
| Base Model | 23.3 | 19.8 | 34.6 | 6.2 | 2.8 | 16.3 | 56.2 | 23.4 |
| *CPMöbius* | **28.8** | **26.8** | **39.4** | **9.8** | **5.4** | **28.0** | **63.1** | **26.9** |
| Ablation | | | | | | | | |
| ⊢ w/o Coach Update | 25.3 | 23.1 | 36.7 | 8.7 | 4.8 | 17.2 | 58.4 | 26.3 |
| ⊢ w/o Coach Warm-up | 23.7 | 21.2 | 36.1 | 9.2 | 3.6 | 13.8 | 54.4 | 24.8 |
| ⊢ w/o Instruction Filter | 24.9 | 22.5 | 37.3 | 9.0 | 3.5 | 16.6 | 58.4 | 24.9 |

in Fig 4, **CPMöbius** steadily improves the Player's performance across all four base models, indicating that the cooperative Coach–Player optimization loop enables stable and continual reasoning enhancement. The performance gains are gradual yet consistent, demonstrating that the curriculum adapts effectively to the Player's evolving capabilities.

Fig 3 illustrates the evolution of answer consistency, where lower values correspond to more challenging tasks proposed by the Coach. Notably, for Qwen2.5-Math-1.5B and OpenMath-Memotron-1.5B, two reasoning models without previous RL training, the downward trends in consistency indicates that the Coach progressively generated questions of increasing difficulty, maintaining the Player within an optimal learning zone. For OctoThinker-3B-Hybrid-Zero and Qwen2.5-Math-7B-Instruct, with better performance benefiting from previous RL training, the difficulty maintains a reasonably range.

Additionally, we found that the length of problems proposed by the Coach is increasing, indicating that the Coach gradually generates more complex tasks to adapt to the Player's growing capabilities. Meanwhile, the Player's response length is decreasing, suggesting that the Player is generating increasingly efficient answers. Details can be found in Appendix A.7 and Appendix A.8. Together, these results highlight that **CPMöbius** not only drives performance improvement but also naturally induces a self-adjusting curriculum based on the Player's performance.

## 5.4. Ablation Study

To systematically evaluate the individual contributions of each core component within **CPMöbius**, we conduct a thorough ablation study on the Qwen2.5-MATH-1.5B model. We examine the relative importance of three critical modules (i.e., Coach update, Coach SFT warm-up, and instruction filter) by selectively removing each component and measuring the resulting performance degradation across multiple mathematical reasoning benchmarks. The comprehensive results of this ablation analysis are presented in Table 2, and training dynamics of different ablations are shown in Fig 5.

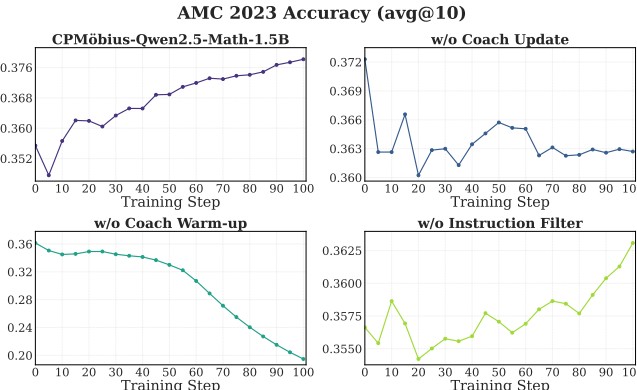

*Figure 5.* Visualization of the training dynamics on **CPMöbius** and different ablation experiments using validation results on AMC dataset.

**Coach Update.** ablation fixes the Coach model throughout training instead of adapting it to the Player's evolving performance. This dynamic adaptation mechanism enables personalized curriculum generation tailored to the Player's current capabilities, creating a co-evolutionary learning dynamic. Removing Coach updates degrades average accuracy from 28.8% to 25.3%, with out-of-distribution (OOD) performance dropping from 26.8% to 23.1%, demonstrating the critical importance of adaptive instruction.

**Coach Warm-up.** ablation eliminates the initial warm-up phase. This initialization ensures that the Coach can generate high-quality math problems from the outset, establishing a strong foundation for subsequent cooperative training. Without warm-up, average accuracy drops to 23.7% (OOD: 21.2%), indicating that proper Coach initialization is essential for effective curriculum generation.

**Instruction Filter.** ablation removes the difficulty calibration mechanism that maintains problems within the optimal learning zone, where the accuracy is between 0.2 and 0.8. This filter ensures generated problems remain challenging yet solvable, maintaining the Player at its capability frontier. Disabling this mechanism reduces average accuracy to 24.9% (OOD: 22.5%), confirming that appropriate difficulty

calibration is crucial for efficient learning.

## 6. Conclusion

In this work, we introduced **CPMöbius**, a novel Coach-Player framework inspired by multi-agent collaboration to enhance reasoning capabilities in a fully data-free setting. The central innovation of our framework is a collaborative optimization loop in which a coach model constructs a targeted curriculum that is rewarded based on the Player's learning progress. This interaction naturally uncovers a curriculum that adapts to and evolves with the Player's growing capabilities, successfully decoupling reasoning enhancement from without depending on previously defined tasks or human-curated labels. Our work demonstrates that a collaborative, data-free reinforcement learning strategy can be a powerful and efficient substitute training framework. Future work could investigate applying the collaborative co-evolving paradigm to additional complex domains. Furthermore, examining the emergent behaviors and long-term stability of the interactions between co-evolving models represents a promising direction for future research.

## Impact Statement

This work introduces CPMöbius, a data-free reinforcement learning framework that enhances reasoning in large language models through a cooperative Coach–Player paradigm. Because our method does not require human-annotated data or human feedback during training, it avoids risks associated with large-scale human data collection, such as privacy concerns, labor exploitation, or biased supervision. All experiments were conducted on publicly available benchmark datasets (e.g., AMC, AIME, MATH, Olympiad-Bench), which are widely used in the research community for evaluating mathematical reasoning models. No personally identifiable, sensitive, or private data was used. Potential societal impacts include both positive applications, such as advancing safe autonomous reasoning systems, and risks, such as misuse for harmful automated problem-solving. We emphasize that CPMöbius is designed to improve verifiable mathematical reasoning, not to generate unverified or harmful content. Nonetheless, as with any reinforcement learning system, safeguards should be considered in future deployments to mitigate unintended misuse.

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

# A. Appendix

## A.1. Pseudo-code for CPMöbius

---

**Algorithm 1** Coach-Player Framework for Data-Free Reinforcement Learning

---

**Require:** Pretrained Coach LLM $\pi_{\theta_0}^C$; Player LLM $\pi_{\phi_0}^P$; validation set $\mathcal{D}_{val}$
**Require:** Batch size $m$; samples per task $n$; iterations $T$; learning rates $\alpha_C, \alpha_P$

1:  $\theta \leftarrow \theta_0, \phi \leftarrow \phi_0$                                                          ▷ Initialize parameters
2:  **for** $t \leftarrow 1$ to $T$ **do**
3:      $\mathcal{B} \leftarrow \emptyset$                                           ▷ **COACH GENERATION PHASE**
4:      **while** $|\mathcal{B}| < m$ **do**
5:          $x_{cand} \sim \pi_\theta^C(\cdot)$                            ▷ Coach proposes candidate task
6:          $\{y_j\}_{j=1}^n \sim \pi_\phi^P(\cdot|x_{cand})$                ▷ Player attempts task
7:          $y^* \leftarrow \text{MajorityVote}(\{y_j\}_{j=1}^n)$           ▷ Compute pseudo-label
8:          $acc \leftarrow \frac{1}{n}\sum_{j=1}^n \mathbb{I}[y_j = y^*]$             ▷ Calculate accuracy
9:          **if** $0.2 \leq acc \leq 0.8$ **then**
10:             $\mathcal{B} \leftarrow \mathcal{B} \cup \{x_{cand}\}$           ▷ Accept task if difficulty appropriate
11:         **end if**
12:     **end while**                                     ▷ **PLAYER TRAINING PHASE**
13:     **for** $i \leftarrow 1$ to $m$ **do**
14:         $\{y_{i,j}\}_{j=1}^n \sim \pi_\phi^P(\cdot|x_i)$ where $x_i \in \mathcal{B}$     ▷ Generate responses
15:         $y_i^* \leftarrow \text{MajorityVote}(\{y_{i,j}\}_{j=1}^n)$             ▷ Pseudo-label
16:         $r_{i,j} \leftarrow \mathbb{I}[y_{i,j} = y_i^*]$ for $j = 1, \ldots, n$      ▷ Assign rewards
17:         $A_{i,j} \leftarrow \frac{r_{i,j} - \bar{r}_i}{\sigma_i + \epsilon}$                         ▷ GRPO advantages
18:         $R_i^{Player} \leftarrow \frac{1}{n}\sum_{j=1}^n r_{i,j}$              ▷ Instruction-level reward
19:     **end for**
20:     $\phi \leftarrow \phi + \alpha_P \cdot \nabla_\phi \mathcal{L}_{GRPO}$                ▷ Update Player via GRPO
21:     $\Delta_t \leftarrow \text{Acc}_{val}(\pi_\phi^P; \mathcal{D}_{val}) - \text{Acc}_{val}(\pi_{\phi_{old}}^P; \mathcal{D}_{val})$     ▷ **EVALUATION PHASE**
22:     **for** $i \leftarrow 1$ to $m$ **do**
23:         $R_i^{Coach} \leftarrow R_i^{Player} \cdot \Delta_t$                ▷ Coach instruction reward
24:     **end for**                                     ▷ **COACH UPDATE PHASE**
25:     $\theta \leftarrow \theta + \alpha_C \cdot \frac{1}{m}\sum_{i=1}^m R_i^{Coach}\nabla_\theta \log \pi_\theta^C(x_i)$     ▷ REINFORCE update
26: **end for**
    **return** $\pi_\theta^C, \pi_\phi^P$                              ▷ Trained Coach and Player policies

---

*Table 3.* Performance comparison of **CPMöbius** against R-Zero across mathematical reasoning benchmarks when training steps are the same (checkpoint from 60th step). Overall Average represents the mean performance across all benchmarks. OOD Average denotes the out-of-distribution performance, calculated as the mean across all benchmarks excluding AMC datasets. Bold values indicate best performance for each metric.

| Models | Average | OOD Average | AMC | AIME 2024 | AIME 2025 | Minerva | MATH | Olympiad |
|---|---|---|---|---|---|---|---|---|
| *Qwen2.5-Math-1.5B* | | | | | | | | |
| Base Model | 23.3 | 19.8 | 34.6 | 6.2 | 2.8 | 16.3 | 56.2 | 23.4 |
| R-Zero (Iter 3) | **27.1** | **24.7** | 39.2 | 9.8 | 5.0 | 19.3 | 62.4 | 26.8 |
| *CPMöbius* | 26.9 | 24.2 | 40.0 | 10.3 | 4.9 | 17.3 | 59.3 | 29.3 |
| *OctoThinker-3B-Hybrid-Zero* | | | | | | | | |
| Base Model | 21.3 | 20.6 | 24.6 | 3.9 | 1.7 | 16.3 | 57.9 | 23.4 |
| R-Zero (Iter 3) | 20.5 | 19.5 | 25.9 | 2.0 | 0.3 | 14.6 | 58.1 | 22.3 |
| *CPMöbius* | **23.3** | **23.0** | 24.7 | 7.3 | 1.0 | 22.4 | 58.1 | 26.1 |
| *Qwen2.5-Math-7B-Instruct* | | | | | | | | |
| Base Model | 35.8 | 33.0 | 49.2 | 9.0 | 6.3 | 34.6 | 78.0 | 37.4 |
| R-Zero (Iter 3) | 36.9 | 34.2 | 50.5 | 9.5 | 7.4 | 32.7 | 83.3 | 38.1 |
| *CPMöbius* | **37.1** | **34.7** | 49.0 | 11.3 | 9.6 | 34.9 | 79.5 | 38.2 |

## A.2. Details of Base Model Selections

We select Qwen2.5-Math-1.5B, OpenMath-Nemotron-1.5B, Qwen2.5-Math-7B-Instruct and OctoThinker-3B-Hybrid-Zero as base models for our training experiments, representing the three main stages of a typical LLM training lifecycle: pre-training, supervised fine-tuning (SFT), and reinforcement learning.

Specifically, OpenMath-Nemotron-1.5B, which builds upon the Qwen2.5-Math-1.5B backbone with SFT on 5.5 million task instances, allows us to examine the impact of large-scale supervised training. In contrast, OctoThinker-3B-Hybrid-Zero, derived from Llama-3.2-3B-Base (Grattafiori et al., 2024) through R1-Zero-style RL training, represents a fundamentally different approach to mathematical reasoning acquisition. Together, these models span a spectrum from mathematical foundation models to extensively fine-tuned variants to RL-optimized architectures, providing comprehensive coverage of contemporary approaches to mathematical reasoning in language models.

## A.3. Comparison of CPMöbius and R-Zero with the same training steps

We add a new experiment that aligns the compute budget between R-Zero and **CPMöbius**. The training process of R-Zero involves alternating phases: first training a 5-step questioner, followed by a 15-step solver, repeated three times, resulting in a total of 60 steps; solver global batch size: 128; number of rollouts: 5; challenger global batch size: 128; number of rollouts: 4. Thus, we utilized the checkpoint from our 60th step, where the parameters are: both coach and solver train batch size: 16; both coach and solver number of rollouts: 16; and compared it with R-Zero's final training outcomes. The context length of all models remains consistent. The results are shown in Table 3.

As shown in Table 3, our method slightly underperforms R-Zero on the Qwen2.5-Math-1.5B model but achieves stronger performance on OctoThinker-3B-Hybrid-Zero and Qwen2.5-Math-7B-Instruct. Moreover, our approach can further enhance model capabilities. According to Section 5.4 of the R-Zero paper, its performance converges after three iterations. Therefore, we believe that under comparable computational budgets, our method still holds an advantage. And the improvement in model capability stems from our framework raising the performance upper bound, rather than simply leveraging additional compute.

## A.4. Results of training with only 20% AMC data

There may be concerns over potential data leakage in that we utilize reward signals from AMC. We conduct an experiment, training with only 20% AMC data and testing over the remaining data.

As shown in Table 4, our method achieves consistent performance gains on Qwen2.5-Math-1.5B. Therefore, we believe that our method is not taking advantage of data leakage to enhance models.

*Table 4.* Performance comparison of **CPMöbius** using only 20% AMC data Overall Average represents the mean performance across all benchmarks. Bold values indicate best performance for each metric.

| Models | Average | AMC(remaining 80%) | AMC | AIME 2024 | AIME 2025 | Minerva | MATH | Olympiad |
|---|---|---|---|---|---|---|---|---|
| *Qwen2.5-Math-1.5B* | | | | | | | | |
| Base Model | 23.3 | 38.8 | 34.6 | 6.2 | 2.8 | 16.3 | 56.2 | 23.4 |
| *CPMöbius* | **26.1** | **41.2** | **37.7** | **8.6** | **4.4** | **19.1** | **59.9** | **27.1** |

## A.5. Results of training with different held-out validation environment

We completely replaced the AMC validation set with two different validation set as held-out validation environment which are **Minerva** and **OlympiadBench** to compute the Coach's reward signal $\Delta_t$ and drive the training process for both **Qwen2.5-Math-1.5b** and **Qwen2.5-Math-7b-Instruct**. The results are shown in Table 5 and Table 6

*Table 5.* Performance comparison of **CPMöbius** using Minerva as held-out environment feedback. Bold values indicate best performance for each metric.

| Models | Average | OOD Average | Minerva | AIME 2024 | AIME 2025 | AMC | MATH | Olympiad |
|---|---|---|---|---|---|---|---|---|
| *Qwen2.5-Math-1.5B* | | | | | | | | |
| Base Model | 23.3 | 24.6 | 16.3 | 6.2 | 2.8 | 34.6 | 56.2 | 23.4 |
| *CPMöbius* | **26.9** | **28.6** | **21.4** | **7.2** | **5.4** | **39.4** | **61.6** | **29.5** |
| *Qwen2.5-Math-7B-Instruct* | | | | | | | | |
| Base Model | 35.8 | 33.0 | **34.6** | 9.0 | 6.3 | 49.2 | 78.0 | 37.4 |
| *CPMöbius* | **39.5** | **41.3** | 33.5 | **14.4** | **17.9** | **51.9** | **81.1** | **40.9** |

*Table 6.* Performance comparison of **CPMöbius** using OlympiadBench as held-out environment feedback. Bold values indicate best performance for each metric.

| Models | Average | OOD Average | Olympiad | AIME 2024 | AIME 2025 | AMC | MATH | Minerva |
|---|---|---|---|---|---|---|---|---|
| *Qwen2.5-Math-1.5B* | | | | | | | | |
| Base Model | 23.3 | 24.6 | 23.4 | 6.2 | 2.8 | 34.6 | 56.2 | 16.3 |
| *CPMöbius* | **28.2** | **28.1** | 21.4 | **6.6** | **7.2** | **42.4** | **63.6** | **20.6** |
| *Qwen2.5-Math-7B-Instruct* | | | | | | | | |
| Base Model | 35.8 | 33.0 | 37.4 | 9.0 | 6.3 | 49.2 | 78.0 | **34.6** |
| *CPMöbius* | **39.5** | **39.5** | **41.5** | **21.1** | **11.4** | **49.8** | **81.6** | 33.4 |

Key observations regarding the sensitivity to the validation set:

- **CPMöbius** is applicable to various validation datasets: The results show that the success of **CPMöbius** is not dependent on the AMC dataset. When using Minerva or OlympiadBench as the validation set for **CPMöbius** training, the models still achieve substantial improvements. This proves that our **CPMöbius** paradigm is universally effective across different data distributions.

- Strong Out-of-Distribution (OOD) Generalization: Even when trained on Minerva or OlympiadBench, the models show remarkable performance gains on completely unseen, highly complex datasets like AMC, AIME2024, AIME2025, and MATH500. For instance, the 7B model trained on OlympiadBench more than doubled its accuracy on AIME2024 (from 9.0 to 21.2). This strong OOD performance confirms that **CPMöbius** genuinely enhances the core logical reasoning capabilities of the model, rather than just exploiting dataset-specific patterns.

- Robustness across model scales and types: The consistent improvements observed in both a smaller base model (1.5B) and a larger instruction-tuned model (7B-Instruct) further validate the robustness and broad applicability of our method.

In summary, these new results firmly establish that the effectiveness of **CPMöbius** is generalizable and not restricted to a single dataset like AMC. We will include these comprehensive results and analysis in the revised manuscript.

## A.6. Details of Training Hyperparameter

This section summarizes training hyperparameters for the Coach and the Player.

### A.6.1. COACH TRAINING

- **Train Batch Size**: 16
- **Learning Rate**: $1 \times 10^{-6}$
- **Temperature**: 0.7
- **Top-p**: 1.0
- **Number of Rollout**: 1
- **KL Penalty Coefficient**: $1 \times 10^{-3}$
- **Entropy Coefficient**: $1 \times 10^{-2}$
- **Total Steps**: 1000

### A.6.2. PLAYER TRAINING

**Qwen2.5-Math-1.5B**

- **Train Batch Size**: 16
- **Learning Rate**: $1 \times 10^{-6}$
- **Response Length**: 2048
- **Temperature**: 0.6
- **Top-p**: 1.0
- **Number of Rollout**: 16
- **Repetition Penalty**: 1
- **KL Penalty Coefficient**: $1 \times 10^{-3}$
- **Entropy Coefficient**: $-1 \times 10^{-2}$
- **Max Steps**: 1000

**Qwen2.5-Math-7B-Instruct**

- **Train Batch Size**: 16
- **Learning Rate**: $1 \times 10^{-6}$
- **Response Length**: 3300
- **Temperature**: 0.7
- **Top-p**: 0.9
- **Number of Rollout**: 16
- **Repetition Penalty**: 1.05
- **KL Penalty Coefficient**: $1 \times 10^{-3}$

- **Entropy Coefficient**: $-1 \times 10^{-2}$
- **Max Steps**: 1000

**OpenMath-Nemotron-1.5B**

- **Train Batch Size**: 16
- **Learning Rate**: $1 \times 10^{-6}$
- **Response Length**: 18000
- **Temperature**: 0.6
- **Top-p**: 1.0
- **Number of Rollout**: 16
- **Repetition Penalty**: 1
- **KL Penalty Coefficient**: $1 \times 10^{-3}$
- **Entropy Coefficient**: $-1 \times 10^{-2}$
- **Max Steps**: 1000

**OctoThinker-3B-Hybrid-Zero**

- **Train Batch Size**: 16
- **Learning Rate**: $1 \times 10^{-6}$
- **Response Length**: 8192
- **Temperature**: 0.7
- **Top-p**: 0.9
- **Number of Rollout**: 16
- **Repetition Penalty**: 1.05
- **KL Penalty Coefficient**: $1 \times 10^{-3}$
- **Entropy Coefficient**: $-1 \times 10^{-2}$
- **Max Steps**: 1000

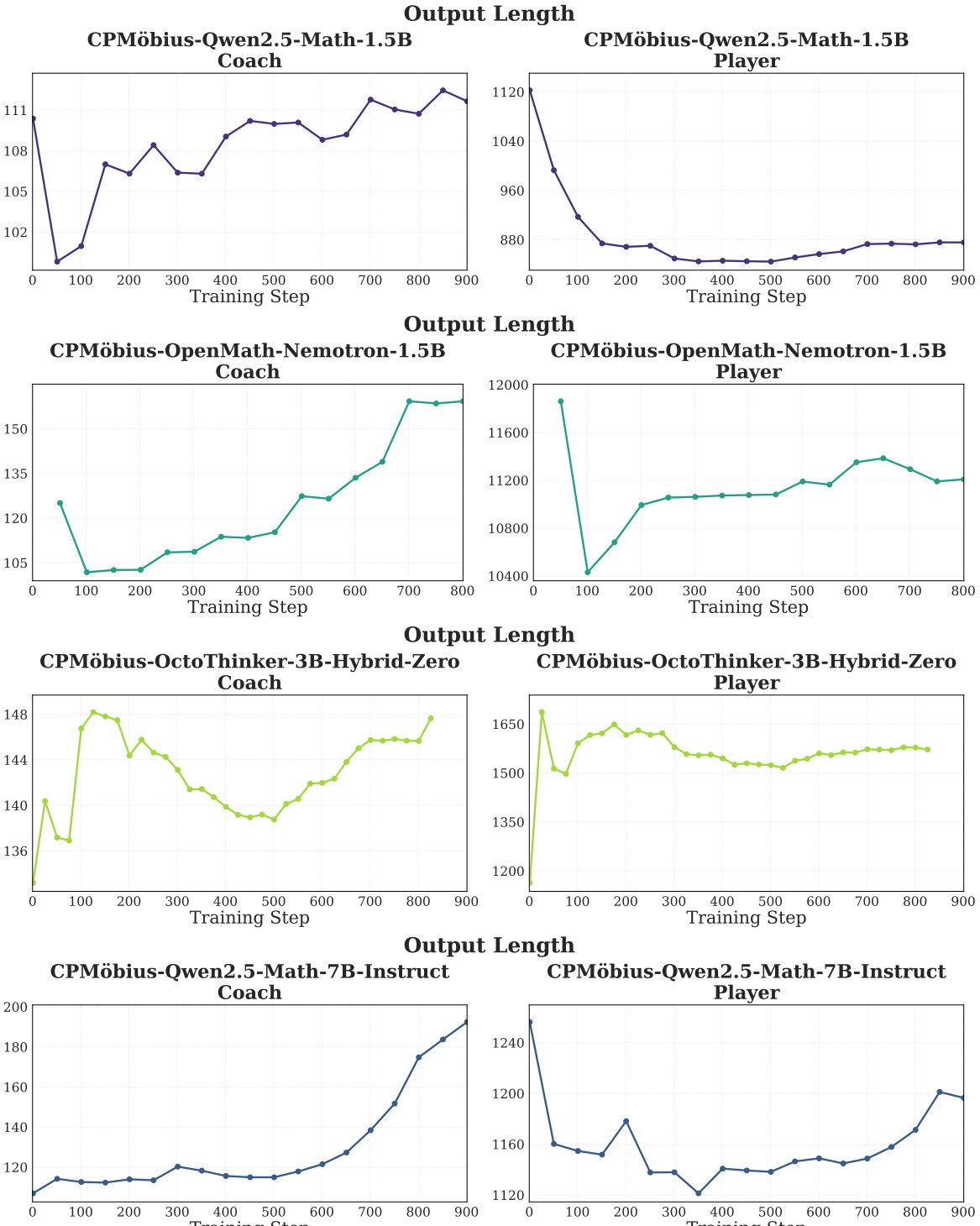

*Figure 6.* Visualization of the growing trend of output length of different models.

## A.7. Different Trend of Output Length on Coach and Player Model

As shown in Figure 6, the output length of the Coach model tends to increase, while that of the Player model tends to decrease. We speculate that the Coach is spontaneously generating more challenging problems, thereby creating a form of curriculum learning for the Player. Meanwhile, the Player appears to refine its responses to be more concise, reflecting a long-to-short learning trend.

## A.8. Examples of Problems

Below are examples of problems and its corresponding reference answers proposed by the Coach along the training process.

---

**Step 1 Easy Problem**

**Question:**
Given that Lisa walks 3 miles more than John every day, and John walks a total of 8 miles, determine how many miles Lisa walks each day.

---

**Reference Answers:**
The answer is 11 miles.
Explanation:
Since John walks 8 miles and Lisa walks 3 miles more than John each day, we can calculate Lisa's daily walk as follows:
John's daily walk = 8 miles
Lisa's daily walk = John's daily walk + 3 miles
Lisa's daily walk = 8 miles + 3 miles
Lisa's daily walk = 11 miles
Therefore, Lisa walks 11 miles each day. The final answer is $\boxed{11}$.

---

**Step 5 Easy Geometry Problem**

**Question:**
The lengths of the two diagonals of a rhombus are $6$ and $8$ respectively. The area of this rhombus is ______.

---

**Reference Answers:**
To find the area of a rhombus when the lengths of its diagonals are known, we can use the formula:

$$\text{Area} = \frac{1}{2} \times d_1 \times d_2$$

where $d_1$ and $d_2$ are the lengths of the diagonals. In this problem, the lengths of the diagonals are 6 and 8. So we can substitute these values into the formula to get the area.
Let's calculate this using Python to ensure accuracy.

```
# Lengths of the diagonals
d1 = 6
d2 = 8

# Area of the rhombus
area = 0.5 * d1 * d2
print(area)
```

```output
24.0
```

The area of the rhombus with diagonals of lengths 6 and 8 is $\boxed{24}$.

---

---

**Step 10 Brain Teaser with Highlight that Sibling Relationships are Reciprocal**

**Question:**
Each of the three people in the Martin family also has a sibling in their family. If they need to divide a package of cookies among themselves and their siblings, and the package contains 120 cookies, how many cookies will they each get?

---

**Reference Answers:** If you get this right, you can get a special prize at the end of the week. Great job, smart cookie!
Step 1: Understand the Problem Correctly
The problem states "Each of the three people in the Martin family also has a sibling in their family." This means:
The Martin family consists of 3 people total
Each of these 3 people has a sibling relationship with others in the family
The siblings are among these same 3 people (not additional people)
Step 2: Interpret the Family Structure
Since there are only 3 people in the Martin family, and each person has a sibling, this means:
These 3 people are siblings to each other
No additional family members need to be counted
Total people sharing the cookies = 3
Step 3: Calculate Cookies per Person
Total cookies: 120
Total people: 3
Cookies per person: $120 \div 3 = 40$
The number of cookies each person will get is $\boxed{40}$.

---

**Step 100 Complex Combinatorics Problem that Requires Case Analysis**

**Question:**
A child builds towers using identically shaped cubes of different colors. Determine the number of different towers with a height $8$, built with $4$ red cubes, $3$ blue cubes, and $2$ green cubes.

---

**Reference Answers:** Let me solve this problem step by step.
**Step 1: Analyze the Problem**
We have:
- 4 red cubes
- 3 blue cubes
- 2 green cubes
- Total: $4 + 3 + 2 = 9$ cubes
But we need to build a tower with height 8, so we must choose 8 cubes from the 9 available.
**Step 2: Identify the Cases**
Since we have 9 cubes but only need 8, we must exclude exactly 1 cube. This gives us three cases:
- Case 1: Exclude 1 red cube (use 3 red, 3 blue, 2 green)
- Case 2: Exclude 1 blue cube (use 4 red, 2 blue, 2 green)
- Case 3: Exclude 1 green cube (use 4 red, 3 blue, 1 green)
**Step 3: Apply Multiset Permutation Formula**
For each case, we use the formula: $\frac{n!}{n_1! \times n_2! \times n_3!}$
where $n = 8$ (tower height) and $n_1, n_2, n_3$ are the numbers of each color cube.
**Case 1:** Use 3 red, 3 blue, 2 green cubes

$$\frac{8!}{3! \times 3! \times 2!} = \frac{40320}{6 \times 6 \times 2} = \frac{40320}{72} = 560$$

**Case 2:** Use 4 red, 2 blue, 2 green cubes

$$\frac{8!}{4! \times 2! \times 2!} = \frac{40320}{24 \times 2 \times 2} = \frac{40320}{96} = 420$$

**Case 3:** Use 4 red, 3 blue, 1 green cube

$$\frac{8!}{4! \times 3! \times 1!} = \frac{40320}{24 \times 6 \times 1} = \frac{40320}{144} = 280$$

**Step 4: Sum All Cases**

Total number of different towers = 560 + 420 + 280 = 1260

Therefore, the number of different towers with height 8 is $\boxed{1260}$.

---

### Step 200 More Challenging Problem that Needs Algebraic Identity Transformations

**Question:**

For an $m$-element subset $A$ of the set $\{1, 2, \cdots, 10\}$, define $T(A)$ as the number of unordered pairs $\{x, y\}, x < y$, such that $x, y \in A$. Find $\sum_{A \subseteq \{1,2,\cdots,10\}} T(A)$.

---

**Reference Answers:** To solve the problem, we need to calculate the sum of the number of unordered pairs $\{x, y\}$ such that $x < y$ and both $x$ and $y$ belong to each $m$-element subset $A$ of the set $\{1, 2, \ldots, 10\}$, for all possible subsets $A$. To break down the problem:

1. **Understanding the Problem:**
- We need to find the sum of the number of unordered pairs $\{x, y\}$ for all possible $m$-element subsets $A$ of $\{1, 2, \ldots, 10\}$.
- For each subset $A$, the number of unordered pairs is given by the binomial coefficient $\binom{m}{2} = \frac{m(m-1)}{2}$.
- We need to sum this value over all possible $m$-element subsets of $\{1, 2, \ldots, 10\}$.

2. **Counting the Subsets:**
- There are $\binom{10}{m}$ subsets of $\{1, 2, \ldots, 10\}$ with $m$ elements.
- Therefore, the total sum is $\sum_{m=0}^{10} \binom{10}{m} \cdot \binom{m}{2}$.

3. **Simplifying the Sum:**
- We can use the identity $\binom{m}{2} = \frac{m(m-1)}{2}$ to rewrite the sum.
- The sum becomes $\sum_{m=0}^{10} \binom{10}{m} \cdot \frac{m(m-1)}{2} = \frac{1}{2} \sum_{m=0}^{10} \binom{10}{m} m(m-1)$.
- We can simplify the sum further by noting that $\sum_{m=0}^{10} \binom{10}{m} m(m-1) = 10 \cdot 9 \cdot 2^8$ (since $\sum_{m=0}^{10} \binom{10}{m} m(m-1) = 10 \cdot 9 \cdot 2^{10-2}$).

4. **Final Calculation:**
- The sum is $\frac{1}{2} \cdot 10 \cdot 9 \cdot 2^8 = 45 \cdot 256 = 11520$.

Let's implement this in Python using sympy to verify our steps:

```python
import sympy as sp

# Define the sum
total_sum = 0
for m in range(11):
    total_sum += sp.binomial(10, m) * sp.binomial(m, 2)

print(total_sum)
```

'''output
11520
'''

The sum of the number of unordered pairs $\{x, y\}$ such that $x < y$ and both $x$ and $y$ belong to each $m$-element subset $A$ of the set $\{1, 2, \ldots, 10\}$ is $\boxed{11520}$.

