# OpenReview forum: "CPMöbius: Iterative Coach–Player Reasoning for Data-Free Reinforcement Learning"
_ICML.cc/2026/Conference — ICML 2026 regular_

### Official Review · Reviewer_QGwh · 2026-02-28

**Soundness:** 3
**Presentation:** 3
**Significance:** 3
**Originality:** 2
**Overall Recommendation:** 4
**Confidence:** 3

**Summary:**

This paper proposes a novel RL-based post-training method that does not require any data for direct model training. Despite certain limitations, the method introduces a novel paradigm that is meaningful for future research. Experimental results also demonstrate the efficacy of the proposed approach.

**Compliance With Llm Reviewing Policy:**

Affirmed.

**Final Justification:**

The added experiments sufficiecntly prove the efficiency and superior performnace of the proposed method. I have no concern about this paper. Therefore I increased my score.

**Key Questions For Authors:**

The design of the coach's reward function is unclear. Why is it based on the player's average reward (estimated accuracy) when GRPO typically relies on group-relative rewards? The paper should provide a clearer explanation or intuition for why this formulation works, beyond the empirical results.

**Limitations:**

Please refer to the weakness and questions.

**Strengths And Weaknesses:**

Strength:
1. This paper proposes a novel data-free paradigm for LLM post-training, where the model can be trained on self-generated questions.
2. Experimental results demonstrate the promising performance of the proposed method, outperforming several strong baselines.

Weakness:
1. Although the authors claim the proposed method is data-free, it still requires a validation set, making this claim somewhat overstated.
2. The experiments are conducted only on the AMC dataset and validated on a few other math-related datasets, which is insufficient to demonstrate the general effectiveness of the proposed method.
3. The proposed method requires both a coach and a player, introducing additional computation and training costs due to the coach. Adding further discussion or analysis on this overhead would be beneficial.

---

> ### Author Rebuttal · Authors · 2026-03-31
>
> **W1**
>
> We agree that our wording should be more precise. Our intended claim is no external training data or no human-curated training curriculum for direct policy learning. In CPMobius, the Player is not trained on external supervised task-solution pairs, instead, it trained from Coach generated tasks and pseudo-labels. The fixed validation set is only used as a environment feedfack for coach to measure whether the newly generated curriculum improves the Player through
>
> $$
> \Delta_t = Acc_{val}(\pi_{\phi_{t+1}}^{P}) - Acc_{val}(\pi_{\phi_t}^{P})
> $$
>
> The Player model never sees the instruction, ground truth, or specific error modes of the AMC problems during training. The Coach only receives a scalar signal that is an accuracy fluctuation to decide how to update the student. This mechanism is similar to standard validation-based model selection or early stopping, where a validation set only influences the final model choice without being part of the training data backpropagation. Therefore, whatever Coach or Player, both of them cannot memorize AMC solutions or exist hacking the metric through parameter optimization on AMC samples.
> We will find a better description to avoid overstating this point.
>
> **W2**
>
> We agree that demonstrating the generalization of our proposed CPMobius method across diverse datasets is crucial.
> To directly address your concern and prove that CPMobius does not merely applicable to certain dataset, we applied CPMobius using Minerva and OlympiadBench to guide the training of two different models. The new experimental results are summarized below:
>
> | Model |  Average | AMC | AIME24 | AIME25 |  Minerva |  MATH500 | OlympiadBench |
> | --- | ---: | ---: | ----: | ----: | ----: | ----: | -----: |
> | Qwen2.5-Math-1.5b      |     23.3 |   34.6 |    6.2 |    2.8 |   16.3 |   56.2 |     23.4 |
> | CPMobius trained on Minerva  |   26.9 |   39.3 |    5.4 |    4.0 | **21.4** |  61.6 |    **29.5** |
> | CPMobius trained on OlympiadBench | **28.2** | **42.4** |  **6.6** |  **7.2** |   20.6 | **63.6** |          28.9 |
>
> | Model |  Average | AMC | AIME24 | AIME25 |  Minerva |  MATH500 | OlympiadBench |
> | --- | ---: | ---: | ----: | ----: | ----: | ----: | -----: |
> | Qwen2.5-Math-7b-Instruct    |     35.8 |     49.2 |      9.0 |      6.3 | **34.6** |   78.0 |    37.4 |
> | CPMobius trained on Minerva    | **39.5** | **51.9** |     14.4 | **17.9** |  30.5 |  81.1 |  40.9 |
> | CPMobius trained on OlympiadBench |     39.4 |     49.8 | **21.2** |  11.4 |     30.9 | **81.6** |      **41.5** |
>
> Key Takeaways:
>
> 1. CPMöbius is applicable to various validation datasets: Using Minerva or OlympiadBench as the validation set for CPMöbius training, the models still achieve improvements.
>
> 2. Strong OOD Generalization: The models show remarkable performance gains on OOD datasets.
>
> In summary, these new results establish that the effectiveness of CPMöbius can generalize.
>
> **W3**
>
> To provide a clearer picture, we conducted a detailed throughput analysis comparing CPMöbius with baselines: R-Zero and Rent-RL.
>
> | | CPMobius | R-Zero | RENT-RL |
> |---|---|---|---|
> | **Qwen2.5-Math-7b-Instruct** | | | |
> | Total Time | 103h (1000 steps) | 40h (60 steps) | 36h (480 steps) |
> | Time per 60 steps | 6.18h | 40h | 4.5h |
> | GPUs | 8 | 8 | 8 |
> | Batch Size | 16×16 | 128×4 (Coach) / 128×8 (Solver) | 40×5 |
> | Total Samples | 256,000 | 320,000 | 19,200 |
> | Generated Problems | 16000 | 32000 | - |
> We found:
>
> 1. **Higher Data Efficiency than R-Zero:** This is because R-Zero relies on a "sampling-then-filtering" strategy, requiring the generation of approximately 8,000 questions to retain only 1,000 for training, introducing a massive hidden overhead.
>
> 2. **Comparable Overhead with Rent-RL but Superior Performance:** Compared to Rent-RL, CPMöbius maintains a similar throughput. This confirms that the inclusion of the Coach module does not bottleneck the system.
>
> **Q1**
>
> We thank the reviewer for this comment. In our framework, the Coach is not optimized with a GRPO-style group-relative advantage. Instead, following Sec. 4.1, the Coach acts as an adaptive curriculum designer and is updated with REINFORCE using instruction-level rewards. The term $R_i^{Player}$ is the average training reward over multiple rollouts for a given instruction, which serves as a stable estimate of how learnable and constructive that the instruction is for the current Player. This is then multiplied by $\Delta_t$, the Player’s global accuracy improvement on a fixed validation set. Therefore, the Coach is reinforced only when an instruction simultaneously achieve high Player performance during training (high $R_i^{Player}$) and contribute to measurable learning progress (positive $\Delta_t$). This design matches our learner-centered objective: to generate tasks that are learnable yet non-trivial. Emperical results show that the questions generated by the Coach are challenging enough to promote skill development and solvable enough to provide a natural curriculum ramp.

---

> > ### Author Rebuttal · Reviewer_QGwh · 2026-04-02
> >
> > Thank the authors for the response. All my concerns got addressed. I raised my score to 4.

---

### Official Review · Reviewer_7St4 · 2026-03-04

**Soundness:** 2
**Presentation:** 4
**Significance:** 3
**Originality:** 2
**Overall Recommendation:** 4
**Confidence:** 4

**Summary:**

This paper proposes CPMobius, a data-free reinforcement learning framework for improving LLM reasoning through an iterative Coach–Player setup. Instead of relying on external supervised data, the method trains a coach model to generate candidate problems and a player model to learn from solving them. A key idea is that the coach is not rewarded simply for producing difficult tasks, but for generating tasks that lead to measurable improvement of the player on a held-out validation set, making the interaction more teaching-oriented than adversarial.

**Compliance With Llm Reviewing Policy:**

Affirmed.

**Final Justification:**

I would like to thank the authors for their detailed rebuttal. After reviewing the responses and the additional clarifications provided, I find that most of my technical concerns have been addressed.

However, my concern regarding W1 remains. While the authors argue that "instruction diversity" is another bottleneck, I find this to be a bit of a conceptual pivot from the original motivation of reducing RLVR annotation costs. Testing exclusively on mathematical benchmarks—where reliable verifiers are ubiquitous—does not fully demonstrate the method’s efficacy in domains where supervision is truly "expensive" or verifiers are unavailable.

Rather than sidestepping this issue by reframing the motivation, I suggest that the authors directly address these limitations. Explicitly discussing the lack of non-mathematical task evaluation as a formal limitation would provide much-needed transparency regarding the framework's current boundaries.

Despite the remaining concerns over domain breadth. I am willing to raise my score to 4. I encourage the authors to tackle problems that are truly impactful for the community, specifically addressing the challenge of verifiability for RLVR in open-domain settings.

**Key Questions For Authors:**

Please see weakness part.

**Limitations:**

The paper does not adequately discuss the method’s generality beyond mathematical reasoning. The paper’s motivation is framed around reducing the cost of RLVR-style supervision, yet the empirical evaluation is largely restricted to math benchmarks where verification signals are relatively accessible. A more complete limitations discussion should explicitly acknowledge that the current evidence does not establish effectiveness in domains where verifiers are unavailable or where RLVR supervision is truly the bottleneck, and that performance may not transfer to open-ended or non-verifiable tasks.

**Strengths And Weaknesses:**

Strength:

S1: A key strength is the teaching-oriented formulation: using performance improvement on a held-out set as the proposer reward provides a principled signal for whether generated problems actually help the solver learn, rather than merely being difficult or adversarial. This is an interesting and potentially valuable idea, as it aligns the proposer’s objective with measurable capability gains and encourages the generation of useful training instances.

S2: The paper is generally well-structured and clearly written. The method description, and experimental narrative are easy to follow, and the overall framing is communicated in a straightforward manner.

Weakness:

W1 The paper motivates the approach as a way to reduce the cost of RLVR-style annotation. However, the empirical validation focuses exclusively on mathematics benchmarks, where verification signals are relatively accessible and labeling is arguably not the primary bottleneck compared to other domains. Without evidence on tasks where RLVR supervision is genuinely scarce or expensive (e.g., domains lacking reliable verifiers), it is difficult to assess how convincing or general the proposed method is with respect to its stated motivation. While I understand that many recent works in this line evaluate mainly on math and coding, I believe the community would benefit from expanding to settings with more severe annotation scarcity rather than continuing to optimize leaderboard performance on these well-studied domains.

W2 The coach is rewarded using the player’s accuracy gain from the previous iteration (i.e., improvement from iteration t−2 to t−1). It is not clear why this signal provides appropriate credit assignment for training the current coach at iteration t. Concretely, why should the player’s improvement under the previous difficulty distribution be a reliable training signal for guiding the coach to generate the next iteration’s problems (potentially at a different difficulty level)? The temporal mismatch between when the reward is measured and when the coach’s actions are taken seems under-justified.

W3 The method relies on the coach to generate training problems. How does the approach ensure that the generated problems are well-posed and internally consistent? In particular, what safeguards exist if the coach produces questions that are nonsensical, ambiguous, or not self-consistent (i.e., have no coherent solution)? Without a mechanism to verify problem validity, the training signal could be noisy or misleading.

W4 The player’s pseudo-labels are produced via majority voting over multiple rollouts. How do the authors ensure that the majority answer is actually correct? Majority voting can suppress exploration: if the correct solution appears only in a minority of rollouts, the method may reinforce an incorrect but popular answer. This raises concerns about confirmation bias and whether the approach can escape local optima when the “right” reasoning trajectory is initially rare.

W5 Equations (5) and (6) appear to imply that R_play is maximized when all rollouts produce identical answers. If this reading is correct, it seems counterintuitive: maximizing coach's reward by encouraging unanimity could favor overconfident behaviors rather than genuine reasoning improvement, and may conflict with the goal of learning from challenging instances. Could the authors clarify whether this interpretation is correct, and if so, why maximizing agreement is desirable in this setting?

---

> ### Author Rebuttal · Authors · 2026-03-31
>
> Thank you for your detailed review. Below, we respond to your concerns and clarify key aspects of our work.
>
> ---
>
> **W1**
> We would like to clarify our motivation. While it is true that mathematical domains have accessible verification signals, the primary bottleneck in scaling RLVR is not just verification, but the generation of a vast and diverse set of instructions. The key challenge we address is whether LLMs can improve their reasoning capabilities without any external training data not just cheaper data. Mathematics serves as an ideal test precisely because reliable verifiers exist, allowing us to rigorously evaluate whether our cooperative Coach–Player loop genuinely improves reasoning rather than merely fitting noisy signals. In the future, we will explore more paradigms in non-verifiable and open-ended domains.
>
> **W2**
>
> We believe there is a misunderstanding of the update order. First, in round t, the current Coach first proposes a batch of tasks, and then the Player is updated on this batch. Next the coach will get the environment feedback for player's training performance which is $\Delta_t$
>
> Finally, the same batch receives instruction rewards
>
> $$
> R_i^{\text{Coach}} = R_i^{\text{Player}} \cdot \Delta_t
> $$
> and is used to update the Coach via REINFORCE. The objective of Coach is
> $$
>     \nabla_\theta J(\theta) = \frac{1}{m}\sum_{i=1}^m R_i^{\text{Coach}} \nabla_\theta \log \pi_\theta^{\text{C}}(x_i) = \mathbb{E}[R_i^{\text{Coach}} \nabla_\theta \log \pi_\theta^{\text{C}}(x_i)]$
> $$
> So the Coach is trained with the reward of the difference of player's performance for the most related iteration, not with an unrelated previous iteration signal.
>
> **W3**
>
> In our experiments, we rarely observed the coach generating meaningless instructions. The difficulty-filtered batching keeps only tasks with rollout-dependent accuracy in the pedagogical optimal zoon $0.2\leq acc_i\leq 0.8$, discarding degenerate cases via dynamic resampling and coach reward mechanism structurally prevent this from happening. If the coach generates unsolvable or meaningless problems, the player's performance will decline, resulting in a negative reward. If the model consistently fails to generate appropriate problems, it will fail to train on OpenMath-Nemotron-1.5B, similar to R-zero's failure.
> I will show the instructions that coach proposed for steps of 0, 100, 200, 400
>
> Step 0:
> - A line passing through the point (2,4) intersects the parabola  $y^2=8x$  at exactly one point. How many such lines are there?
>
> Step 100:
>
> - In triangle $ABC$, let $\\overline{AB} = 7\\sqrt{2}$, and let the coordinates of points $A$ and $B$ be $(0,0)$ and $(7\\sqrt{2}, 0)$, respectively. Suppose that the internal angle bisector of $\\angle BAC$ intersects line segment $\\overline{BC}$ at point $D$ such that $\\overline{AD} = \\overline{AC}$. Find the length of $\\overline{AC}$.
>
> Step 200:
> - In $\\triangle ABC$, $AC=6$, $\\cos B=\\dfrac{1}{3}$, $C=\\dfrac{\\pi }{4}$.
> (I) Find the value of $\sin A$;
> (II) Find the value of $\overrightarrow{BC} \cdot \overrightarrow{CA}$
>
> Step 400:
> - Find the smallest value of \\(z\\) such that there exists an integer \\(n > 17\\) with \\(n^2 \\equiv z \\pmod{50}\\)
>
> **W4**
>
> The work of TTRL demonstrates that majority voting, a test-time scaling method, is a very efficient and suitable way to promote RL training and reward shaping. Judging from the improvement results of TTRL and Player Model, this method of using pseudo-labels is effective. Just like a lucky hit does not necessarily reinforce the correct one, but it can suppress the wrong one. In many scenarios beyond human capabilities, where the reward model cannot provide the correct reward, and where there is no verifiable reward, majority voting is a relatively efficient way to solve the reward shaping problem. Therefore, we do not need to pay too much attention to whether pseudo-labels are actually correct or wrong. Based on the experimental results, Player model does not get stuck in a bad attractor.
>
> **W5**
>
> First, Even if $R_i^{\text{Player}}$ is high, the task is useful for the Coach only when it also yields positive \Delta_t. Moreover, the Coach filter explicitly removes tasks with $acc_i\geq 0.8$, so trivially easy or high-agreement tasks are excluded. The objective is therefore to favor learnable yet non-trivial tasks. Next, I will clarify it again.
> In our implementation, the Coach reward at iteration $t$ is defined as the difference in Player validation performance:
> $$
>     R_i^{\text{Coach}} = R_i^{\text{Player}} \cdot \Delta_t
> $$
> where $R_i^{\text{Player}} = \frac{1}{n}\sum_{j=1}^n r_{i,j}$ represents the average training reward achieved by the Player on instruction $x_i$, and $\Delta_t$ measures the Player's accuracy improvement on the validation set. This quantity can be positive if the current curriculum improves the Player or negative if the curriculum harms the Player. Like we mentioned at W2.

---

> > ### Author Rebuttal · Reviewer_7St4 · 2026-04-03
> >
> > I would like to thank the authors for their detailed rebuttal. After reviewing the responses and the additional clarifications provided, I find that most of my technical concerns have been addressed.
> >
> > However, my concern regarding W1 remains. While the authors argue that "instruction diversity" is another bottleneck, I find this to be a bit of a conceptual pivot from the original motivation of reducing RLVR annotation costs. Testing exclusively on mathematical benchmarks—where reliable verifiers are ubiquitous—does not fully demonstrate the method’s efficacy in domains where supervision is truly "expensive" or verifiers are unavailable.
> >
> > Rather than sidestepping this issue by reframing the motivation, I suggest that the authors directly address these limitations. Explicitly discussing the lack of non-mathematical task evaluation as a formal limitation would provide much-needed transparency regarding the framework's current boundaries.
> >
> > Despite the remaining concerns over domain breadth. I am willing to raise my score to 4. I encourage the authors to tackle problems that are truly impactful for the community, specifically addressing the challenge of verifiability for RLVR in open-domain settings.

---

### Official Review · Reviewer_AFFi · 2026-03-13

**Soundness:** 3
**Presentation:** 3
**Significance:** 3
**Originality:** 3
**Overall Recommendation:** 5
**Confidence:** 4

**Summary:**

This paper proposes an Iterative Coach-Player Reasoning framework for Data-Free RL. It notes that compared to the adversarial setup of self-play, which is prone to instability and tends to collapse into nonsensical or unlearnable proposed tasks for RL training, their framework adopts a collaborative approach, achieving notable improvements across multiple datasets, including out-of-distribution settings.

**Compliance With Llm Reviewing Policy:**

Affirmed.

**Final Justification:**

Thanks for the detailed response. I hope the authors can incorporate the content from the rebuttal into the revised version of the paper. This is a clearly written and easy-to-follow paper.

**Key Questions For Authors:**

- Could the authors provide a case study under the Self-Play setting, corresponding to the observation in this paper that the proposed questions become increasingly longer and more difficult, to better illustrate the differences between the two approaches and the advantages of the proposed method?

- During GRPO training, the authors adopted the TTRL method. Are there other alternatives here, or is TTRL used as a relatively default and general-purpose method in the unsupervised setting?

- The authors have included sample questions at certain step intervals in the appendix. Could the authors provide the complete prompts (for both the Coach and the Player) to facilitate a better understanding of the implementation details of this framework?

**Limitations:**

See Weakness and Key Questions.

**Strengths And Weaknesses:**

- This paper proposes a novel data-free RL training paradigm that incorporates rewards related to accuracy differences and capability levels, which is easy to understand and highly readable.
- The paper conducts a fairly comprehensive evaluation, covering multiple math datasets and comparing against the R-Zero and RENT baselines.
- For the Player Model, there theoretically exists an optimal capability improvement trajectory corresponding to the Coach's optimal question proposal strategy. This point could benefit from deeper analysis and discussion.
- Within this framework, could the RL process encounter situations where the Player (or Coach) stagnates due to the poor performance of the other party? Could the authors supplement the paper with an analysis of such related failure cases?

---

> ### Author Rebuttal · Authors · 2026-03-31
>
> Thank you for your detailed review. Below, we respond to your concerns and clarify key aspects of our work.
>
> ---
> W1:
> In our early experiments, when using the raw Qwen2.5-Math-7B-Instruct directly as the Coach, we observed that as training progressed, the model would increasingly exhibit unstable task generation behavior, including repetitive outputs, garbled text, and, in some cases, failure to generate valid mathematical problems at all. We also observed a related issue in preliminary tests with OpenMath-Nemotron-1.5B under the R-Zero self-play setting: the model was often unable to reliably generate meaningful questions for the initialized challenger. That is the reason we tried to warm-start the Coach using two math subsets from the PRIME-Eurus-2-SFT-Data dataset. In practice, 4k examples were sufficient to give the Coach a reliable initialization for generating well-formed and diverse math problems, while keeping the warm-up lightweight relative to the full training process. In subsequent experiments, we found that the warm-started Qwen2.5-Math-7B-Instruct rarely showed the earlier failure modes such as repetitive outputs, garbled generations, or inability to propose math questions.
>
> Q1:
> We track the generation of increasingly long and difficult mathematical reasoning questions by R-Zero (Challenger-Solver) and CPMöbius across three iterative self-play stages, with the difficulty designed to align with the edge of the solver/player’s current capability (the core design of both methods). The case fully reflects the distinct question generation characteristics of the two frameworks driven by their different incentive mechanisms:
>
> | Iteration | R-Zero | Steps | CPMobius |
> |---|---|---|---|
> | 1 | *Question:* Solve for x: 3x - 7 = 20. | 0 | *Question:* A line passing through the point (2,4) intersects the parabola at exactly one point. How many such lines are there? |
> | 2 | *Question:* A right triangle has a hypotenuse of 13 cm and one leg of 5 cm; find its area and the height on the hypotenuse. | 200 | *Question:* In triangle $ABC$, let $\overline{AB} = 7\sqrt{2}$, and let the coordinates of points and be and $(7\sqrt{2}, 0)$, respectively. Suppose that the internal angle bisector of intersects line segment at point such that $\overline{AD} = \overline{AC}$. Find the length of $\overline{AC}$. |
> | 3 | *Question:* Prove that for all positive integers n, $1^2+2^2+...+n^2=\frac{n(n+1)(2n+1)}{6}$, then compute the sum of squares from 1 to 50. | 400 | *Question:* Find the equation of the circle that passes through points $A(0,1)$ and $B(1,-2)$, and whose center lies on the line $x+y+5=0$. |
>
> We expect the CPMöbius to be advantageous in scenarios:
> - Environment. When a strong unified executable verifier (code executor) is unavailable, but a held-out progress environment feedback is still available.
> - Task Generation. In AZR, the task generation and solving capabilities are entangled, and proposer and solver share parameters, it's possible to generate extreme tasks in pursuit of high rewards, leading to training fluctuations.
> - Scalability. The decoupled and heterogeneous architecture allows independent scaling of Coach and Player, such as using a smaller size Coach with a larger size Player or using Coach and Player of similar size but from different model families. This flexibility makes heterogeneous scaling much easier than in single-model self-play, where both roles are tied to the same model.
>
> Q2:
>
> For unsupervised RLVR, there are multiple alternatives for reward design. Broadly, they can be categorized into intrinsic reward methods, where rewards are derived from the model’s own signals (e.g., confidence or output consistency), and external reward methods, where rewards come from mechanisms outside the model, such as unlabeled data or computational asymmetry.
> We use TTRL as a practical and relatively general-purpose choice within the family of ensemble-based intrinsic rewards. Its core idea is to generate multiple rollouts and use agreement among the outputs as the reward signal, under the assumption that higher consensus is correlated with higher correctness. This is closely related in spirit to other unsupervised RLVR approaches, such as entropy-minimization-based methods like EMPO [3]; we also compare against RENT-RL as such a baseline.
> We do not claim that TTRL is the only possible choice. Rather, we adopt it because it is particularly efficient and well suited to tasks with well-defined answer formats, such as mathematics and coding, where majority voting and consistency can provide a stable and effective unsupervised learning signal.
>
> Q3:
>
> The Coach uses "Generate a valid math problem." as the base prompt to generate math problems. The Player receives these problems and wraps them with the model's native chat template (e.g., <|im_start|>user\n[problem]\n<|im_end|>\n<|im_start|>assistant) before solving them. We will include the prompts in the appendix and supplementary material in the revision.

---

> > ### Author Rebuttal · Reviewer_AFFi · 2026-04-03
> >
> > In the rebuttal, regarding the weaknesses about "there theoretically exists an optimal capability improvement trajectory" and "the analysis of failure cases caused by one side's poor performance", I feel the responses were not entirely clear to me (possibly because I didn't fully grasp the authors' intent). I will maintain my original rating (weak accept).

---

> > > ### Author Response · Authors · 2026-04-07
> > >
> > > Thank you for the follow-up, and we appreciate your maintaining the weak accept rating. We realize our previous response on these two points may have been too compressed, so I will restate the intended meaning.
> > >
> > > **Theoretically exists an optimal capability improvement trajectory:**
> > > - We do not claim that CPMöbius explicitly solves for a globally optimal curriculum. Our intended point is simpler: the Coach is rewarded by the Player’s held-out improvement $\Delta_t$, so it is encouraged to generate problems that are more useful for the Player’s current learning stage. In practice, when proposed problems are too easy, too hard, or otherwise uninformative, they typically lead to small or non-positive improvement and therefore provide weak training signal to the Coach. In this sense, CPMöbius can be viewed as a practical online approximation to adaptive curriculum learning, without requiring a hand-designed difficulty schedule.
> > >
> > > **Analysis of failure cases**
> > > - To provide a clearer illustration of the failure cases that can arise when one side of the loop performs poorly, we include a concrete example from our baseline experiment of R-Zero on OpenMath-Nemotron-1.5B. In this setup, R-Zero trains the model to generate questions wrapped in ``<question></question>`` tags, so that the generated questions can be extracted and passed to the solver for the next training stage. However, during our reproduction, we found that the trained OpenMath-Nemotron-1.5B challenger frequently failed to follow this required format. Specifically, it often did not wrap its generated questions in the prescribed ``<question></question>`` tags, instead using alternative formats such as “Consider the problem:” or “Problem:”, which could not be properly parsed by the R-Zero pipeline. In addition, it sometimes generated highly repetitive or meaningless content, and occasionally reproduced the prompt template itself verbatim. Outputting strings such as ``<question>\n{The full problem statement on one or more lines}\n</question>\n\n`` rather than an actual question. As a result, the solver could not obtain usable training questions, and the training pipeline failed to proceed effectively. This example illustrates how, in a self-play setting, poor performance on the proposer side can directly block learning on the solver side and lead to system-level stagnation or failure.

---

### Official Review · Reviewer_uM1E · 2026-03-13

**Soundness:** 3
**Presentation:** 3
**Significance:** 3
**Originality:** 2
**Overall Recommendation:** 4
**Confidence:** 3

**Summary:**

This paper proposes CPMöbius, a data-free reinforcement learning framework for improving mathematical reasoning in LLMs. The key idea is a collaborative Coach–Player paradigm: the Coach generates math problems tailored to the Player's current ability, and the Player learns to solve them using majority-voting pseudo-labels and GRPO. Unlike R-Zero, the Coach is rewarded based on the Player's actual learning progress ensures tasks remain in the Player's zone of proximal development. The method is evaluated on four base models across six math benchmarks, showing consistent improvements over baselines like RENT and R-Zero.

**Compliance With Llm Reviewing Policy:**

Affirmed.

**Final Justification:**

The authors' rebuttal reasonably addressed my main concerns and I maintain my recommendation of weak accept.

**Key Questions For Authors:**

Please refer to Weaknesses.

**Limitations:**

The paper does not include an explicit discussion of limitations. Several aspects in Weaknesses deserve further consideration.

**Strengths And Weaknesses:**

Strengths:

1. Novel cooperative paradigm. Instead of adversarial self-play, the Coach is directly incentivized to help the Player improve rather than to challenge it. This shift from competition to cooperation is conceptually interesting and leads to more stable training. The four-stage loop with pseudocode also makes the framework easy to follow and reproduce.

2. Interpretable reward design. The Coach's reward is tied to the Player's actual performance gain on a validation set, offering a clean and meaningful optimization signal. This avoids the vague reward attribution issues common in self-play methods and makes the training objective straightforward to understand and verify.

3. Comprehensive experiments and ablations. The method is tested across multiple base models at different scales and training stages, showing consistent improvements on six math benchmarks, especially on out-of-distribution tasks like Minerva. Ablation studies confirm that each component — Coach updates, difficulty filtering, and Coach warm-up — contributes meaningfully, with clear performance drops when any one is removed.


Weaknesses:

1. Insufficient discussion and comparison with Absolute Zero Reasoner (AZR)

The paper cites Absolute Zero (Zhao et al., 2025) only briefly in the Related Work section, grouping it together with several other data-free methods without any detailed discussion. Given that AZR arguably the most closely related published work in this space, this treatment feels insufficient.

Both CPMöbius and AZR address the same core problem — improving LLM reasoning via self-generated tasks without external data — and share a similar high-level structure. I would suggest the authors add AZR as an experimental baseline if its code is publicly available. If a direct experimental comparison is not feasible, at minimum a detailed qualitative comparison in the Related Work section would significantly strengthen the paper, and under what scenarios the cooperative paradigm is expected to offer advantages over AZR's self-play approach.

2. Sensitivity to validation set choice.

The Coach's reward signal depends on Δt.

Δt​ computed on a fixed AMC validation set, but no experiments explore how different choices of validation se​t affect the final performance. It remains unclear whether the observed gains would hold with a different validation set.

3. Reliance on Coach warm-up data.

Although the "data-free" claim applies to the co-evolution stage, the ablation shows that Coach warm-up is critical (average accuracy drops from 28.8 to 23.7 without it). A brief discussion on how the choice and size of warm-up data influence results would be helpful.

---

> ### Author Rebuttal · Authors · 2026-03-31
>
> Thank you for your detailed review. Below, we respond to your concerns and clarify key aspects of our work.
>
> ---
> W1:
>
> For Absolute Zero's proposer-solver mechanism, it's a self-play between individual models, emphasizing self-evolution through adversarial competition between the proposer and solver. Absolute Zero heavily relies on the code executor as a verifiable environment to provide absolutely objective feedback, including validating self-generated tasks and verify answers for proposer and solver provides the correct answer. This is a very hard constraint. Therefore, Absolute Zero is a framework that relies on the code executor to provide "verifiable rewards" for reinforcement learning training. For CPMöbius's coach-player mechanism, the emphasis is on co-evolution between the coach and player, focusing on collaboration to improve the player model's performance ceiling. CPMöbius doesn't require a hard verifier to provide absolutely objective feedback and validate deterministic rewards. The main role of the AMC validation set is to provide an environment feedback for the coach to judge whether the player has improved. Therefore, we chose R-zero as one of our baselines. Self-play mechanism of R-zero does not rely on verifiable supervision from a hard verifier, making it more suitable as our baseline.
> We expect the cooperative paradigm to be particularly advantageous in scenarios:
> - Environment. When a strong unified executable verifier (such as a code executor) is unavailable, but a held-out progress environment feedback is still available.
> - Task Generation. In AZR, the task generation and solving capabilities are entangled, and proposer and solver share parameters, it's possible to generate extreme tasks in pursuit of high rewards, leading to training fluctuations.
> - Scalability. The decoupled and heterogeneous architecture allows independent scaling of Coach and Player, such as using a smaller size Coach with a larger size Player or using Coach and Player of similar size but from different model families. This flexibility makes heterogeneous scaling much easier than in single-model self-play, where both roles are tied to the same model.
>
> W2 :
>
> To directly address this concern, we conducted new ablation studies. We completely replaced the AMC validation set with two different datasets Minerva and OlympiadBench to compute the Coach's reward signal $\Delta_t$ and drive the training process for both Qwen2.5-Math-1.5b and Qwen2.5-Math-7b-Instruct.
> The results are presented below:
>
> Table 1: Performance of Qwen2.5-Math-1.5b
> | Model | Average | AMC | AIME2024 | AIME2025 | Minerva | MATH500 | OlympiadBench |
> |---|---|---|---|---|---|---|---|
> | Qwen2.5-Math-1.5b | 23.3 | 34.6 | 6.2 | 2.8 | 16.3 | 56.2 | 23.4 |
> | CPMöbius-Minerva | 26.9 | 39.3 | 5.4 | 4.0 | **21.4** | 61.6 | **29.5** |
> | CPMobius-OlympiadBench | **28.2** | **42.4** | **6.6** | **7.2** | 20.6 | **63.6** | 28.9 |
>
> Table 2: Performance of Qwen2.5-Math-7b-Instruct
> | Model | Average | AMC | AIME2024 | AIME2025 | Minerva | MATH500 | OlympiadBench |
> |---|---|---|---|---|---|---|---|
> | Qwen2.5-Math-7b-Instruct | 35.8 | 49.2 | 9.0 | 6.3 | **34.6** | 78.0 | 37.4 |
> | CPMöbius-Minerva  | **39.5** | **51.9** | 14.4 | **17.9** | 30.5 | 81.1 | 40.9 |
> | CPMobius-OlympiadBench | 39.4 | 49.8 | **21.2** | 11.4 | 30.9 | **81.6** | **41.5** |
>
> In summary, these new results firmly establish that  not restricted to a single validation set like AMC. We will include these comprehensive results and analysis in the revised manuscript.
>
> W3:
>
> In our early experiments, when using the raw Qwen2.5-Math-7B-Instruct directly as the Coach, we observed that as training progressed, the model would increasingly exhibit unstable task generation behavior, including repetitive outputs, garbled text, and, in some cases, failure to generate valid mathematical problems at all. We also observed a related issue in preliminary tests with OpenMath-Nemotron-1.5B under the R-Zero self-play setting: the model was often unable to reliably generate meaningful questions for the initialized challenger.
> Under our compute budget, we were constrained to use a relatively small model as the Coach, and Qwen2.5-Math-7B-Instruct provided the best trade-off between capacity and efficiency. To improve its task-proposal ability, we warm-started the Coach using two math-focused subsets from the 230k PRIME-Eurus-2-SFT-Data dataset, namely mathinstruct-math and openmathinstruct-2-augmented_math. We sampled 4k examples from these two subsets for warm-up. We wanted a warm-up set that was large enough to stabilize mathematical question generation. In practice, 4k examples were sufficient to give the Coach a reliable initialization for generating well-formed and diverse math problems. In subsequent experiments, we found that the warm-started Qwen2.5-Math-7B-Instruct rarely showed the earlier failure modes such as repetitive outputs, garbled generations, or inability to propose mathematical questions.

---

> > ### Author Rebuttal · Reviewer_uM1E · 2026-04-01
> >
> > The rebuttal has resolved my major concerns. I hope the authors will incorporate the AZR discussion and the new validation set experiments into the revised paper. I will maintain my positive assessment (weak accept).

---

### Decision · Program_Chairs · 2026-04-30

**Decision:**

Accept (regular)

**Comment:**

This work presents a cooperative Coach–Player paradigm for data-free reinforcement learning of LLM reasoning, where the Coach is rewarded based on the Player's actual learning progress rather than task difficulty alone.

Reviewers found the cooperative formulation conceptually interesting and the experimental results consistently positive across multiple model scales (scores: 5/4/4/4). The teaching-oriented reward design and comprehensive ablations were highlighted as strengths. Primary concerns centered on the "data-free" claim given the reliance on a validation set and Coach warm-up, limited evaluation beyond math benchmarks, and the relationship to Absolute Zero Reasoner. The authors responded with experiments, detailed cost analysis, and clarified the conceptual distinctions from AZR. For the camera-ready, the authors should explicitly acknowledge the math-only evaluation scope as a limitation and refine the "data-free" framing.

I recommend acceptance.